# Cross-Country Adaptation of a Psychological Flexibility Measure: The Comprehensive Assessment of Acceptance and Commitment Therapy Processes

**DOI:** 10.3390/ijerph19063150

**Published:** 2022-03-08

**Authors:** Ambra Mara Giovannetti, Jana Pöttgen, Elisenda Anglada, Rebeca Menéndez, Jürgen Hoyer, Andrea Giordano, Kenneth Ian Pakenham, Ingrid Galán, Alessandra Solari

**Affiliations:** 1Unit of Neuroepidemiology, Fondazione IRCCS Istituto Neurologico Carlo Besta, 20133 Milan, Italy; andrea.giordano@istituto-besta.it (A.G.); alessandra.solari@istituto-besta.it (A.S.); 2Institute of Neuroimmunology and Multiple Sclerosis, University Medical Center Hamburg-Eppendorf, 20246 Hamburg, Germany; j.poettgen@uke.de; 3Department of Neurology, University Medical Center Hamburg-Eppendorf, 20246 Hamburg, Germany; 4Centre d’Esclerosi Múltiple de Catalunya (Cemcat), Hospital Universitari Vall d’Hebron, 08035 Barcelona, Spain; eanglada@cem-cat.org (E.A.); rmenendez@cem-cat.org (R.M.); igalan@cem-cat.org (I.G.); 5Institute of Clinical Psychology and Psychotherapy, Technische Universität Dresden, 01187 Dresden, Germany; juergen.hoyer@tu-dresden.de; 6Department of Psychology, University of Turin, 10124 Turin, Italy; 7School of Psychology, Faculty of Health and Behavioural Sciences, University of Queensland, St. Lucia, QLD 4072, Australia; k.pakenham@psy.uq.edu.au

**Keywords:** cultural adaptation, linguistic validation, outcome measures, CompACT, psychological flexibility

## Abstract

Purpose: The Comprehensive assessment of Acceptance and Commitment Therapy (ACT) processes (CompACT) is a 23-item self-report questionnaire assessing psychological flexibility, which is the overarching construct underpinning the ACT framework. We conducted a two-phase project to develop validated versions of the CompACT in three languages: phase 1—cross-cultural adaptation; and phase 2—psychometric validation of the questionnaire for use in Italy, Germany and Spain. This article focuses on the first phase. Methods: We translated and culturally adapted the CompACT in the three target languages, following the ISPOR TCA Task Force guidelines. The process was overseen by a translation panel (three translators, at least two multiple sclerosis (MS) researchers and a lay person), ACT experts and clinicians from the research team of each country and the original CompACT developers. We debriefed the new questionnaire versions via face-to-face interviews with a minimum of four adults from the general population (GP) and four adults with MS in each country. Results: The translation-adaptation process went smoothly in the three countries, with some items (7 in Italy, 4 in Germany, 6 in Spain) revised after feedback from ACT experts. Cognitive debriefing showed that the CompACT was deemed easy to understand and score in each target country by both GP and MS adults. Conclusions: The Italian, German and Spanish versions of the CompACT have semantic, conceptual and normative equivalence to the original scale and good content validity. Our findings are informative for researchers adapting the CompACT and other self-reported outcome measures into multiple languages and cultures.

## 1. Introduction

Acceptance and Commitment Therapy (ACT) is a transdiagnostic, empirically based third generation cognitive behavioral therapy approach that conceptualizes psychological suffering primarily as a function of experiential avoidance and values-inconsistent behavior [1]. ACT aims to promote psychological flexibility, a cornerstone of mental health and resilience [2,3], by targeting six positive psychological skills, which constitute the psychological flexibility framework underpinning ACT: (1) acceptance—openness to experience; (2) cognitive defusion—observing thoughts rather than taking them literally; (3) present moment awareness—mindfulness; (4) self-as-context—contact with a sense of self that is continuous and provides flexible perspective taking; (5) values—freely chosen personally meaningful life directions; (6) committed action—values-guided effective action [4].

Each skill has been shown to be related to better mental health, lower risk of disease, better health outcomes for those already diagnosed with illness, and neurobiological resilience factors [5]. ACT has a growing evidence base for improving psychosocial outcomes across a range of health conditions, including psychiatric or organic chronic conditions [6]. Gloster et al. [6] found that ACT is more effective than waitlist and placebo conditions and at least as effective as the traditional cognitive behavioral therapies. In line with these findings, two recent studies showed that psychological flexibility is an important mental health protective factor during COVID-19 pandemic. It mitigates the detrimental impacts of COVID-19 risk factors [7] and health anxiety [8] on mental health. These findings support the importance of ACT-based large-scale public interventions that target psychological flexibility.

Developing and disseminating (self-report) measures of psychological flexibility that are acceptable, reliable and sensitive to change is crucial. Equally important is having these scales available in different languages, to allow consistent and comparable use of this outcome measure across countries for clinical and research purposes, and the development of international trials of ACT-based interventions.

The Comprehensive Assessment of ACT processes (CompACT) is a global measure of psychological flexibility [9]. The authors developed this scale to overcome the limitations of the most widely used measure of psychological flexibility, the Acceptance and Action Questionnaire-II (AAQ-II) [10]. One weakness of the AAQ-II is that instead of measuring psychological flexibility, it assesses psychological inflexibility and is used as a proxy for psychological flexibility [11,12]. In addition, Wolgast [13] has argued that the AAQ-II is confounded with distress-outcome variables, and Francis et al. [9] argued that the AAQ-II is limited in its assessment of the six psychological flexibility processes due to a preponderance of items that focus on acceptance/experiential avoidance and defusion/fusion processes, neglecting other important processes (i.e., values and committed action, contact with the present moment and self-as-context) within the ACT model.

The CompACT consists of 23 items that are rated on a 0–6 Likert scale and are grouped into three subscales: Openness to Experience (10 items), Behavioral Awareness (five items) and Valued Action (eight items). This three-factor structure reflects Hayes et al.’s [14] definition of psychological flexibility in terms of three “dyadic” processes (each consisting of two of the six aforementioned psychological skills): (1) “openness to experience and detachment from literality” (acceptance; cognitive defusion); (2) “self-awareness and perspective taking” (present moment awareness; self-as-context); and (3) “motivation and activation” (values; committed action).

A total score is calculated as the sum of the three subscale scores (range 0–138), with higher values indicating greater psychological flexibility. The English version of the CompACT demonstrated good internal consistency, and converged and diverged in theory-consistent ways with other measured variables, that is, higher levels of psychological flexibility were associated with lower levels of distress and higher levels of health and wellbeing [9].

At the commencement of the present study, another measure of psychological flexibility, the Multidimensional Psychological Flexibility Inventory [15] was published in English. Although this scale shows promise as a measure of psychological flexibility, it is lengthy (60 items) and potentially less practical for use in clinical and research settings. The CompACT was chosen because of its brevity and demonstrated sound psychometric properties [9].

The present project was developed by an international team of clinicians and researchers who are experts in the field of multiple sclerosis (MS) and ACT-based intervention. Some of these experts have published studies on the application of ACT-based intervention for people with MS in their respective countries [3,16,17,18]. We believed the first step in laying the foundation for designing and conducting international trials of ACT-based interventions is to validate the CompACT, as a measure of the core ACT construct (psychological flexibility), in three languages in a concerted effort to enhance comparability and cross-cultural utility of the scale. Due to our interest in MS, we designed the study so that it not only focused on people from the general population, but also people with MS. 

The project involved two phases: (1) translation-adaptation of the CompACT in standard Italian, German and Spanish for use in the general population (GP) and for adults with MS; and (2) psychometric validation of the scale in people with MS across the three countries. This article focuses on phase 1.

## 2. Materials and Methods

The study was conducted in three research centers, each representing a different European country/language: (1) Fondazione IRCCS Istituto Neurologico Carlo Besta, Italy (coordinating center); (2) Department of Neurology at the University Medical Center Hamburg Eppendorf, Germany; (3) Centre d’Esclerosi Múltiple de Catalunya—Hospital Universitari Vall d’Hebron, Barcelona, Spain. The study was undertaken in accordance with the recommendations of the Declaration of Helsinki. The protocol received ethical clearance from the ethics committees of the coordinating center (12/09/2018, internal ref: 54; first amendment approved 12/12/2018, internal ref: 57; second amendment approved 08/05/2019, internal ref: 62) and participating centers (23/04/2019, clearance number: PV6040 for Germany; 01/03/2019, clearance number: PR(AG)29/2019 for Spain). All participants gave written informed consent. In each country, the CompACT [9] was translated and culturally adapted from the original UK English into the target language (Italian, German or Spanish), following the ISPOR TCA Task Force guidelines [19].

### 2.1. Preparation

The study processes were centralized at the Unit of Neuroepidemiology, Fondazione IRCCS Istituto Neurologico Carlo Besta (referred to as the methodology hub, MH from hereon). The MH had responsibility for the overall methodology and cross-country coordination. The MH sought permission from the CompACT authors and involved them in the study, devised the materials and procedures, trained the center’s principal investigators, and provided oversight throughout the project.

### 2.2. Translation-Adaptation

In each country, the translation-adaptation of the CompACT consisted of four phases which are summarized in Figure 1.

In phase 1, two qualified translators whose mother tongue was in the target language and who had fluency in UK English, produced two independent forward translations. The translators were specifically instructed to use a colloquial style that was easy to understand by the GP. A translation panel consisting of the two translators, at least two MS researchers and a lay person reviewed the forward translations (Panel Meeting 1) and a reconciled translation was produced.

In phase 2, the reconciled translation was independently translated back into UK English by a third qualified translator, whose mother tongue was UK English and who was fluent in the target language. The backward translation was produced without access to the original CompACT and without consulting the other translators.

The translation panel plus the backward translator compared the back translation with the original (Panel Meeting 2). When necessary: CompACT translations, back translations and reconciled translations were scrutinized, and queries were presented to CompACT authors. The reconciled translation was further revised if necessary. This process resulted in an advanced translation (Advanced Version 1).

In phase 3 (expert feedback), an ACT expert whose mother tongue was in the target language was appointed to read the Advanced Version 1 and provide the translation panel with comments and feedback on the accuracy and theoretical coherency of the translated questionnaire. Feedback from CompACT authors was also obtained. Because the CompACT authors did not speak any of the three target languages, they received the three translation grids (see below) and the back translations and were asked to compare the original CompACT with the back translation produced by each of the three translation panels, and to identify any inconsistencies and conceptual errors.

Meetings 1 and 2 were audio-recorded, and the whole process was reported in a translation grid (Appendix A), which was made available to each panel member to facilitate discussion. Differences were resolved by consensus. Challenging statements, uncertainties and rationale for final decisions were reported in the translation grid. The grid also contained queries for the ACT expert and for the CompACT authors. After Meeting 2, the translation grid was reviewed by each panel member for validation.

Meeting 3 (teleconference) took place after feedback from the ACT expert and CompACT authors was obtained. It involved the translation panel plus the backward translator, who considered each suggested change and query. This process resulted in an advanced translation (Advanced Version 2).

Phase 4: After Meeting 3, the new advanced translation (Advanced Version 2) was used during cognitive debriefing and a final version (Final Translation) was produced.

### 2.3. Cognitive Debriefing

In each country, cognitive debriefing involved the GP and MS participants.

#### 2.3.1. Sampling and Recruitment

Participants were selected using a purposeful sampling technique ensuring diversity in age and education. Participants were adults (age ≥ 18 years), fluent in the language of the participating country.

MS patients had a confirmed diagnosis [20] and no severe cognitive impairment (clinical judgement), which precluded communication. The study was advertised via social media platforms and within the personal networks of the researchers involved. A dedicated researcher in each participating center received expressions of interest, informed potential participants about the study, checked the eligibility criteria, and provided the informed consent form.

#### 2.3.2. Interview Procedure for Cognitive Debriefing

In each country, a minimum of four interviews were planned for each group (GP and MS); sampling ended when no new content emerged [21]. At each center, the face-to-face interviews were conducted by one trained psychologist/physician in a quiet room. The interviewer checked that all eligibility criteria were met and then invited the participant to provide his/her demographic information (age, sex, education) on a standard form. Clinical information (Expanded Disability Status Scale score [22], MS type, and duration) was obtained from the referring neurologist.

First, the participant was asked to complete the CompACT questionnaire. Then, the interviewer asked a series of open-ended questions to explore the interviewee’s understanding of the questionnaire as a whole, the introduction, each item, and the response options. This investigation was based on an interview guide designed by the investigators (Appendix A) [23]. Finally, the interviewee was asked about the questionnaire’s length, layout, and readability. The interviewer took written notes (interviews were not recorded).

### 2.4. Final Translations

Each translation was proofread locally and at the MH, to ensure that any minor errors were corrected, and the layout was similar, before psychometric testing.

### 2.5. Analyses

Continuous data were summarized using means, standard deviations, medians and ranges/interquartile ranges, while categorical data were described in terms of frequencies.

At each participating center, interview notes were reviewed using content analysis to identify areas of misunderstanding and where modifications to wording or layout were indicated [21,23]. Reports were compared and discussed jointly by the three interviewers and the principal investigator, and then by the MH.

## 3. Results

### 3.1. Translations

The demographic characteristics of the three translation panels are reported in Table 1.

#### 3.1.1. Italy

The two Italian forward translations were discussed with the translation panel members, and no relevant discrepancies were found, and a reconciled translation was produced. The backward translation was consistent with the original English version. One query was produced by the panel: item 7, “stressful” was translated as “costa fatica” (“takes effort”) as in Italian “stressante” was considered too general. The CompACT authors accepted the reconciled version 2 (Appendix A).

Two Italian ACT experts evaluated the quality (coherency with ACT theory) of the advanced Italian version 1. They suggested changes to five verbs (items 1–3, 22, 23), one noun (item 15), and one sentence structure (item 21). The translation panel discussed each suggestion and refined the seven items (Appendix A).

#### 3.1.2. Germany

The two German forward translations were discussed with the members of the translation panel. In 10 cases, differences (all minor) were discussed and a reconciled version was produced (Appendix A). The backward translation was consistent with the original English version. The backward translation was in line with the original English version and the CompACT authors did not raise any inaccuracies.

The German ACT expert suggested the following minor changes: three verbs (items 1, 6, 12) and a preposition (item 18; Appendix A). The translation panel discussed all the comments received, agreed on the changes to items 1, 6 and 18, and refined item 12 (Appendix A).

#### 3.1.3. Spain

The two Spanish forward translations were discussed with the translation panel members, and no relevant discrepancies were found, and a reconciled version was produced. The only exception was item 6, where the translation panel was uncertain on the translation of “get so caught up”. They discussed this issue with the ACT expert who suggested they use “absorto” (“absorbed”) instead of “enfrascado” (“immersed”). The backward translation was in line with the original English version and the CompACT authors did not raise any issues.

The Spanish ACT expert suggested changes to four verbs (items 6, 7, 13, 14) and two adjectives (items 8, 10). The translation panel discussed each suggestion and refined all six items (Appendix A).

The MH compared the back translations of the three language versions with each other and the source version. No discrepancies were found.

### 3.2. Cognitive Debriefing

#### 3.2.1. GP Sample

Thirteen people from the GP participated in the debriefing process (Italy = 5; Germany = 4; Spain = 4). Most (69%) were female, and the mean age was 50 years (min–max = 32–72). Most (85%) participants had a high school diploma or a degree.

The majority of the sample found the questionnaire (i.e., instructions, items, response scale and layout) clear and accessible: “*I found the questionnaire clear and concise, the instructions are also clear, and I can’t see any problem in using the response scale*” (Spain, GP). However, six items were judged unclear by at least one participant: item 3 (one German and one Spanish participant), item 4 (one German and two Italian participants), item 6 (one German and one Spanish participant), item 13 (one Italian participant), item 20 (one Spanish and two Italian participants) and item 22 (one Italian participant). With the exception of one Spanish participant, the response options were easily understood and judged to be appropriate by all. Interestingly, an Italian participant suggested that examples be provided to improve the clarity of items. Details on the problematic aspects of the questionnaire identified by participants are reported in Table 2 (quotes are reported in Appendix A).

#### 3.2.2. MS Patients

Eighteen people with MS participated in the debriefing process; most (67%) were female, and the mean age was 44 years with the Italian sample having the youngest mean age. Most (94%) participants had a high school diploma or a degree. The mean MS duration was 11 years, with the Spanish subgroup having the longest mean duration. As expected, most participants (61%) had relapsing-remitting MS. The MS severity range was wide, with Expanded Disability Status Scale scores ranging from 1 to 8, of a possible score range between 0 (no impairment) and 10 (death due to MS) [22]. Sociodemographic and clinical characteristics of participants are reported in Table 3.

Most people with MS found the questionnaire clear and accessible: “This questionnaire is really interesting. It made me think about myself in different situations… Sometimes I needed time to answer because I am not used to thinking about these things, but items were clear” (Italy, people with MS). The only items that were judged unclear were item 2 (one Italian and two German participants), item 5 (one German participant), and item 20 (three Italian participants). Spanish participants had no concerns with any items, but they provided general comments on the questionnaire: two of them noted difficulties with the scoring system because they found the response gradient vague. Two participants reported that most items were very long and mentioned that this could be a problem for people with MS with cognitive impairment. Details on the problematic aspects of the questionnaire identified by participants are reported in Table 2 (quotes are reported in Appendix A).

#### 3.2.3. MH Discussion of the Findings

The MH sifted the findings of the cognitive debriefing in both samples, in order to check whether any of the issues raised were related to the conceptual equivalence between the source and target versions.

In the GP sample, concerns were due to the abstractness of the underlying construct: committed action (item 3); avoidance of internal experience (item 4); acceptance (item 13 and 22); defusion (item 20). In addition, some GP participants found the item unclear because they included more than one ACT process, or their inflexibility counterpart (e.g., cognitive fusion and inaction, item 6). One participant suggested that providing an example could help clarify item meaning. Although a potentially useful suggestion, the fact that the issue was raised by only one participant was deemed not sufficient to change the questionnaire.

In the MS sample, the CompACT was deemed easy to understand and score in each target country. Specifically, the concerns raised on 3/23 items reflect the abstractness of the underlying constructs: acceptance (item 2); committed action (item 5); cognitive defusion (item 20). The two concerns on the response options (gradient not clear; inclusion of a “neutral” option) pertain to the structure of the original scale and will inform the subsequent study phase (i.e., psychometric testing), as item-response-theory techniques are required for assessing item ‘scalability’ of translated tools [24]. Finally, the concern about lengthy items and how this may be problematic for people with severe cognitive impairment was only relevant to specific subgroups; therefore, no further revisions of the target questionnaires were proposed.

The final versions of the CompACT in each of the three languages are reported in the Appendix A.

## 4. Discussion

In this article, we describe the simultaneous adaptation across three European languages of a self-report measure of psychological flexibility that was developed in the UK (The CompACT questionnaire). We reported on the coordinated translational techniques, procedures and challenges in achieving equivalence of the three language versions of the CompACT. This constitutes the first phase of the larger project designed to develop validated versions of the CompACT in three languages. The psychometric testing of these CompACT versions will be reported in a subsequent article.

Our adaptation framework was based on the ISPOR TCA Task Force guidelines [19]. We aimed to reconcile a rigorous and shared methodology with the limited resources that are typical of independent research; the overall budget for the CompACT for MS project was of EUR 15,000, mostly allocated to fees for the nine qualified translators. Some guidelines include more than one backward translation, or a (forward and backward) translation panel [25]. In order to ensure that the three CompACT versions are sensitive to local contextual variations while remaining equivalent to the original measure, we involved the CompACT authors, as well as ACT experts from each participating country. This involvement was key to assuring not only the equivalence (semantic, conceptual, and normative) of the source and target versions, but also the capacity of the measure to capture the meaning of a latent construct in another population [19]. Some changes in wording were recommended by the ACT experts whose mother tongue was in the target language and who could therefore identify nuances in the meaning of item wording that deviated from core ACT theory. In addition, an accurate cognitive debriefing process was included, as recommended by Epstein and colleagues [26]. These authors valued the use of a cognitive debriefing process as a crucial step to enhance the quality of the translation process [26].

Some of the issues raised from the transcultural adaptation process were not sufficient to warrant revisions; specifically, the reordering of response options, reducing the length of the scale, and the addition of examples due to the abstractness of some questions. However, a robust and well-traced translation-adaptation process is a prerequisite for subsequent psychometric testing [27].

We believe that the quality checking of the translation of self-report outcome measures is often neglected, and yet it deserves the same rigorous standards as psychometric validation. It is anticipated that the procedures and resources documented in this article will be useful in future research endeavours designed to adapt the CompACT to other languages and cultures.

### Limitations

The MH assured a consistency of materials and procedures; however, it is possible that the quality of the data and procedures differed across the countries. For example, Italy used a team of qualified translators with longstanding collaboration with the MH. Furthermore, most participants in the cognitive debriefing had at least a high school diploma and lower levels of education were underrepresented. In the next project phase, we will assess the psychometric validity of the scale on a large population of MS patients (paper in preparation), and further investigate its cross-country equivalence.

## 5. Conclusions

A high quality cross-country linguistic validation of the CompACT was successfully performed in three languages: Italian, German and Spanish. Therefore, the first reliable and valid measure of psychological flexibility, the CompACT, is now available for all countries that have one of these languages as their national language. This first step will enable the collection of important data on psychological flexibility and evaluation of the efficacy of psychological flexibility-based interventions in at least three European countries. Moreover, this article offers a step-by-step guide on how to perform a cross-country validation of a questionnaire at an international level that involves participants with different cultures and languages. The methodology used can be easily applied to other contexts/countries and different health conditions.

## Figures and Tables

**Figure 1 ijerph-19-03150-f001:**
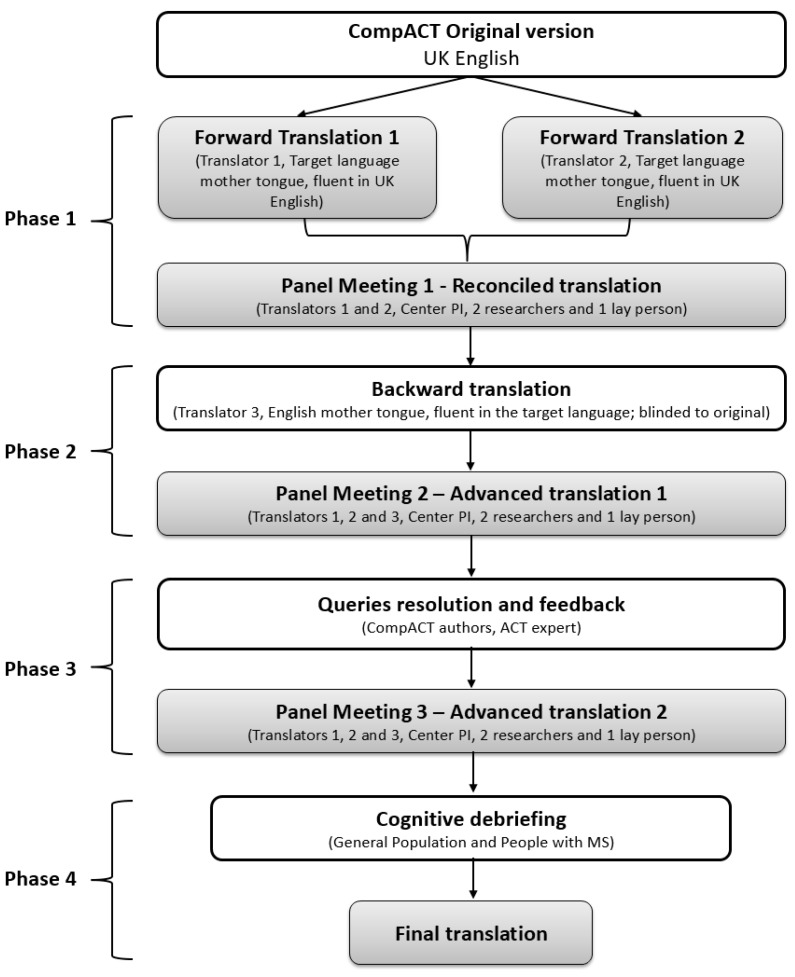
Translation process. MS is multiple sclerosis; PI is principal investigator; ACT is acceptance and commitment therapy.

**Table 1 ijerph-19-03150-t001:** Characteristics of the translation panels.

Characteristic	Italy (n = 8)	Germany (n = 8)	Spain (n = 8)
Age, years *	57.0 (34.0–65.0)	48.5 (24.0–62.0)	53.5 (42.0–68.0)
Women **	7 (87.5)	5 (62.5)	7 (87.5)
Profession **			
*Translator/conference Interpreter*	3 (37.5)	3 (37.5)	3 (37.5)
*Neurologist*	1 (12.5)	1 (12.5)	-
*Physician*	-	1 (12.5)	1 (12.5)
*Psychologist/Neuropsychologist*	1 (12.5)	1 (12.5)	2 (25.5)
*Lay person (technician)*	1 (12.5)	-	-
*Lay person (administration employee)*		1 (12.5)	1 (12.5)
*ACT expert*	2 (25.5)	1 (12.5)	1 (12.5)
Place of residence **			
*Milan (Italy)*	6 (75.0)	-	-
*Enna (Italy)*	1 (12.5)	-	-
*Parma (Italy)*	1 (12.5)		
*Hamburg (Germany)*	-	7 (87.5)	-
*Dresden (Germany)*	-	1 (12.5)	-
*Barcelona (Spain)*	-	-	5 (62.5)
*Madrid (Spain)*	-	-	1 (12.5)
*Almeria (Spain)*			1 (12.5)
*Esperança (Portugal)*			1 (12.5)

* Median (min-max); ** (n, %).

**Table 2 ijerph-19-03150-t002:** Results of the content analysis: concerns about the questionnaire.

Where in the Questionnaire	Content of Concern	Italy		Germany		Spain	
		*GP*	*PwMS*	*GP*	*PwMS*	*GP*	*PwMS*
Item 2—One of my big goals is to be free from painful emotions	- *Item too vague*		1		1		
	- *Unclear expression: “big goal”*				1		
Item 3—I rush through meaningful activities without being really attentive to them	- *Unclear item*					1	
	- *Unclear expression: “meaningful activities”*			1			
Item 4—I try to stay busy to keep thoughts or feelings from coming	- *Unclear item* - *Unclear expressions: “thoughts” and “feelings”*	2		1			
Item 5—I act in ways that are consistent with how I wish to live my life	- *Item too vague*				1		
Item 6—I get so caught up in my thoughts that I am unable to do the things that I most want to do	- *Unclear item, too nested*			1		1	
Item 13—I am willing to fully experience whatever thoughts, feelings and sensations come up for me, without trying to change or defend against them	- *Unclear expression: “to fully experience”.*	1					
Item 20—Thoughts are just thoughts–they don’t control what I do	- *Unclear item*		1			1	
	- *Unclear expression: “thoughts do not control what I do”*		1				
	- *Unclear expression: “thoughts are just thoughts”*	2	1				
Item 22—I can take thoughts and feelings as they come, without attempting to control or avoid them	- *Unclear expression: “take thoughts and feeling as they come”.*	1					
Response options	- *Difficult to understand the response gradient* - *Option 3 (“Neither agree nor disagree”) is not useful*		1			1	2
	- *Use number instead of label*					1	
General comments	- *Too long statements (items)*						2
	- *Unclear item, too nested*			1		1	
	- *Concerns on possible comprehension difficulties for people with cognitive impairment*						2
	- *Identical items*	1	2		1	1	
	- *Include examples to clarify items*	1					

Note: GP = general population; PwMS = people with multiple sclerosis. Numbers indicated the frequency of each concern.

**Table 3 ijerph-19-03150-t003:** Clinical and demographic data of the people with multiple sclerosis who participated in the cognitive debriefing.

Characteristic	Italy (n = 8)	Germany (n = 6)	Spain (n = 4)
Age, years *	36.5 (23.0–55.0)	47.0 (36.0–55.0)	53.5 (41.0–66.0)
Women **	6 (75.0)	3 (50.0)	3 (75.0)
Education **			
*Middle school diploma*	0 (0)	1 (17)	0 (0)
*High school diploma*	5 (62.5)	2 (33)	2 (50)
*Degree*	3 (37.5)	3 (50)	2 (50)
Disease duration, years *	5.0 (2.0–34.0)	8.5 (1.0–21.0)	15.0 (7.0–23.0)
MS type **			
*Relapsing-remitting*	6 (75.0)	4 (66.7)	1 (25.0)
*Secondary-progressive*	1 (12.5)	2 (33.3)	2 (50.0)
*Primary-progressive*	1 (12.5)	0	1 (25.0)
EDSS score *	3.0 (1.0–8.0)	2.5 (1.0–6.5)	5.0 (4.0–6.5)

* Median (min-max); ** (n, %).

## Data Availability

The data presented in this study are available in the main manuscript and in Appendix A “Audit Trail”.

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
