# Peer review of "Cross-Country Adaptation of a Psychological Flexibility Measure: The Comprehensive Assessment of Acceptance and Commitment Therapy Processes"

_ijerph, 2022, doi:10.3390/ijerph19063150_

Round 1

Reviewer 1 Report

This paper presents a study that aimed to validate thee different versions (Italian, German, and Spanish) of a scale to measure psychological flexibility: the Comprehensive assessment of ACT (Acceptance and Commitment Therapy) processes - CompACT. The study is well conducted.

Suggestions and questions (answers can/should be used to improve the paper):
1. If there is only one subheading in section 1 (1.1.), it should be removed.
2. "This article focuses on phase 1" - Is it focused on phase 1 because phase 2 is not concluded? Or did the authors decide to divide the findings?
3. Do people speak Spanish in Esperança (Portugal)? I am not against anything about this, but this could be declared in the manuscript.
4. Are the study limitations a threat to its validity? This needs to be discussed.
5. Future work could be declared.

Specific comments:
- All abbreviations should be declared at the first occurrence (e.g., NSW, AAR, SPIRIT, etc.)
- Remove the final dot from the title.
- i.e. -> i.e., (add comma)
- Some abbreviations (MS and ACT) in the caption of the figure 1 can be removed. They are presented in the text.
- When possible, avoid short paragraphs.
- I could not get "(Online Source 4)". What is it? Would it be a supplementary material (https://www.mdpi.com/journal/ijerph/instructions#suppmaterials)?
- Are "Limitations" and "Limitations" subheadings of the discussion section? So, they are 4.1. and 4.2., respectively.

Author Response

REVIEWER 1

This paper presents a study that aimed to validate three different versions (Italian, German, and Spanish) of a scale to measure psychological flexibility: the Comprehensive assessment of ACT (Acceptance and Commitment Therapy) processes - CompACT. The study is well conducted.

Suggestions and questions (answers can/should be used to improve the paper):

  1. If there is only one subheading in section 1 (1.1.), it should be removed.

We deleted the subheading 1.1.

  1. "This article focuses on phase 1" - Is it focused on phase 1 because phase 2 is not concluded? Or did the authors decide to divide the findings?

We purposely divided the findings into two papers. Putting together the two phases of this three-country project would have compressed findings. See also in this regard the Discussion (page 12): ‘We believe that the quality checking of the translation of self-report outcome measures is often neglected, and yet it deserves the same rigorous standards as psychometric validation. It is anticipated that the procedures and resources documented in this article will be useful in future research endeavors designed to adapt the CompACT to other languages and cultures’.

  1. Do people speak Spanish in Esperança (Portugal)? I am not against anything about this, but this could be declared in the manuscript.

We have now specified in the Introduction that we translated-adapted the questionnaire in standard Italian (i.e. for Italy), Spanish (i.e. for Spain), and German (i.e. for Germany). We did not do that in the original manuscript as this specification is used when the translation is to a language other than the standard (e.g. Swiss Italian, Swiss German, or Mexican Spanish).

The modified statement is reported here: “The project involved two phases: 1) translation-adaptation of the CompACT in standard Italian, German, and Spanish for use in the general population (GP) and for adults with MS, and 2) psychometric validation of the scale in people with MS across the three countries. This article focuses on phase 1.”

  1. Are the study limitations a threat to its validity? This needs to be discussed. The two limitations of the study are minor. The training, coordination and monitoring from the Neuroepidemiologic Unit at FINCB, which has established expertise in linguistic validation, prevented the occurrence of major deviations to the study protocol. The medium-high schooling of participants in the cognitive debriefing is unfortunately a common occurrence. In the next project phase, we will assess the psychometric validity of the scale on a large population of MS patients (paper in preparation), and will further investigate its cross-country equivalence. The following sentence has been added on page 12 (Discussion): “In the next project phase, we will assess the psychometric validity of the scale on a large population of MS patients (paper in preparation), and further investigate its cross-country equivalence.”
  2. Future work could be declared.

In response of this reviewer’s point, of point 4 above, and of reviewer 2 point 3, we have added the following sentence on page 12 (Discussion): “In the next project phase, we will assess the psychometric validity of the scale on a large population of MS patients (paper in preparation), and further investigate its cross-country equivalence.”  

Specific comments:

- All abbreviations should be declared at the first occurrence (e.g., NSW, AAR, SPIRIT, etc.)

We checked throughout the manuscript and provided the extended definition of each abbreviation at first occurrence.

- Remove the final dot from the title.

Thank you. We removed the dot from the title.

- i.e. -> i.e., (add comma)

Thank you. We added the comma where missed.

- Some abbreviations (MS and ACT) in the caption of the figure 1 can be removed. They are presented in the text.

Thanks for your comment. If it is not a mistake for you, we would keep the abbreviation as they provide a clearer and immediate understanding to the reader.

- When possible, avoid short paragraphs.

In revising the manuscript, we have avoided short paragraphs where possible. We felt it necessary to retain some concise paragraphs to visually ‘chunk’ information into a readable format. The methodological procedures involving three countries and three research teams adds to the complexity of the study. Hence, we were mindful of the need to present information in the clearest way possible, consequently we have retained some short paragraphs that allow for visual scanning of text to select desired information.

- I could not get "(Online Source 4)". What is it? Would it be a supplementary material (https://www.mdpi.com/journal/ijerph/instructions#suppmaterials)?

We renamed “Online Source” as “Supplementary File”. Changes in the text are in red.

- Are "Limitations" and "Limitations" subheadings of the discussion section? So, they are 4.1. and 4.2., respectively.

Thank you, we modified the subheadings accordingly.

Reviewer 2 Report

This is a study evaluating the translation and adaptation of the CompACT psychological flexibility tool to three different European languages with expert feedback, followed by cognitive debriefing with healthy controls and patients with multiple sclerosis. It offers a good and very analytical approach to ensure accurate and appropriate adaptation of psychological batteries to foreign languages, but delves into too much methodological detail which might be of limited interest to the end-user. 

I could not find a reasoning as to why patients with MS were selected as a population. Can you include a sentence or two in the Introduction explaining why these patients would benefit from ACT? 

I would also be interested in seeing some numerical results or other quantitative data comparing the results of the tests from the different countries. If the participants took the test, why shouldn't we see if the results are comparable between HC and PwMS and between the different language versions? 

I would also like to see some more detail in the next steps of this project at the end of the discussion and why it will be useful going forward to expand the use of CompACT across different countries and for what purpose the data in PwMS will be used. 

Author Response

REVIEWER 2

  1. This is a study evaluating the translation and adaptation of the CompACT psychological flexibility tool to three different European languages with expert feedback, followed by cognitive debriefing with healthy controls and patients with multiple sclerosis. It offers a good and very analytical approach to ensure accurate and appropriate adaptation of psychological batteries to foreign languages, but delves into too much methodological detail, which might be of limited interest to the end-user.

We thank the reviewer for his/her appreciation of the manuscript. We emphasize the importance of using valid and reliable outcome measures to improve science, as far as PROs are concerned, equivalence of such measures across languages and cultures is a key component (and often neglected). The consistent international interpretation and analysis of results is only possible if the data are from ‘one instrument’ (cross-country invariance). The regulatory agencies have voiced the importance of linguistic validation of PROs. EMA (EU now includes 25 countries) did this in 2004. In coherence with this approach, and to sensitize researchers on its importance, we have documented the process in the manuscript. We have purposely submitted this manuscript to an open access journal that allows sufficient airing of contents.

  1. I could not find a reasoning as to why patients with MS were selected as a population. Can you include a sentence or two in the Introduction explaining why these patients would benefit from ACT?

In response to this reviewer, we have added a sentence in the Introduction (page 3), explaining why we included on MS patients: “The present project was developed by an international team of clinicians and researchers who are experts in the field of multiple sclerosis (MS) and ACT-based intervention. Some of these experts have published studies on the application of ACT-based intervention for people with MS in their respective countries [16-19]. We believed the first step in laying the foundation for de-signing and conducting international trials of ACT-based interventions is to validate a the CompACT, as a measure of the core ACT construct (psycho-logical flexibility), in three languages in a concerted effort to enhance com-parability and cross-cultural utility of the scale. Due to our interest in MS, we designed the study so that it not only focused on people from the general population, but also people with MS.

The project involved two phases: 1) translation-adaptation of the Com-pACT in standard Italian, German, and Spanish for use in the general population (GP) and for adults with MS, and 2) psychometric validation of the scale in people with MS across the three countries. This article focuses on phase 1.”

  1. I would also be interested in seeing some numerical results or other quantitative data comparing the results of the tests from the different countries. If the participants took the test, why shouldn't we see if the results are comparable between HC and PwMS and between the different language versions?

The cognitive debriefing is a qualitative method, thus the data obtained are of limited value for any quantitative comparison. Quantitative data on the questionnaire and its psychometric properties will be provided in the manuscript describing Phase 2 results.

  1. I would also like to see some more detail in the next steps of this project at the end of the discussion and why it will be useful going forward to expand the use of CompACT across different countries and for what purpose the data in PwMS will be used.

In response of this reviewer’s point, and that of reviewer 2 point 5, we have added the following sentence on page 12 (Discussion): “In the next project phase, we will assess the psychometric validity of the scale on a large population of MS patients (paper in preparation), and further investigate its cross-country equivalence.” 
